# Event Stream GPT: A Data Pre-processing and Modeling Library for Generative, Pre-trained Transformers over Continuous-time Sequences of Complex Events

**Matthew B. A. McDermott**
Department of Biomedical Informatics
Harvard University
matthew_mcdermott@hms.harvard.edu

**Bret Nestor**
Allen School of Computer Science & Engineering
University of Washington

**Peniel Argaw**
John A. Paulson School of
Engineering and Applied Sciences
Harvard University

**Ye Jin**
Department of Biomedical Informatics
Harvard University

**Isaac Kohane**
Department of Biomedical Informatics
Harvard University

## Abstract

Generative, pre-trained transformers (GPTs, a type of "Foundation Model") have reshaped natural language processing (NLP) through their versatility in diverse downstream tasks. However, their potential extends far beyond NLP. This paper provides a software utility to help realize this potential, extending the applicability of GPTs to continuous-time sequences of complex events with internal dependencies, such as medical record datasets. Despite their potential, the adoption of foundation models in these domains has been hampered by the lack of suitable tools for model construction and evaluation. To bridge this gap, we introduce Event Stream GPT (ESGPT), an open source library designed to streamline the end-to-end process for building GPTs for continuous-time event sequences. ESGPT allows users to (1) build flexible, foundation-model scale input datasets by specifying only a minimal configuration file, (2) leverage a Hugging Face compatible modeling API for GPTs over this modality that incorporates intra-event causal dependency structures and autoregressive generation capabilities, and (3) evaluate models via standardized processes that can assess few and even *zero-shot* performance of pre-trained models on user-specified fine-tuning tasks.

## 1   Introduction

The unprecedented performance of large language models (LLMs) in natural language processing (NLP) has led to a new paradigm in machine learning. Instead of being dominated by single-task, supervised learning models, this paradigm is characterized by general-purpose, pre-trained "foundation models." These models, exemplified by the "Generative, Pre-trained Transformer" (GPT) architecture, deliver state-of-the-art performance on diverse downstream tasks in a highly data-efficient manner [24]. The success of GPT models over natural language data is driven in

37th Conference on Neural Information Processing Systems (NeurIPS 2023) Track on Datasets and Benchmarks.

part by the fact that the generative, language modeling forecasting task is "universal" across NLP downstream tasks—meaning any natural language task can be recast as a language modeling task through prompting. However, this property is not unique to NLP, and GPT-equivalent architectures may hold significant promise in other domains as well. In particular, this work explores the application of these models to the domain of continuous-time sequences of complex, multimodal events (event streams), specifically Electronic Health Record (EHR) datasets. In particular, this paper presents a data preprocessing and modeling library that makes it significantly easier to build generative, auto-regressive transformers over event stream data, then to fine-tune or use those models for zero-shot prediction over diverse fine-tuning tasks on these data, such as in predicting the risk of a patient of dying in the hospital or of needing an early readmission to the hospital from structured EHR data.

Two critical barriers inhibit the application of foundation model research to modalities like EHR data. First, these data modalities do not come in a single, unified format, and compelling datasets are often private and unshareable. This hampers foundation model research by requiring dataset-specific preprocessing pipelines and hindering one's ability to assess the model generalizability. Second, modeling continuous-time sequences of complex events is more challenging than modeling ordinal sequences of tokens, as GPTs over these modalities must account for non-ordinal inputs; complex, multi-modal emission distributions; and intra-event causal relationships.

In this paper, we address these gaps with Event Stream GPT (ESGPT), an open source software package[1], API, and evaluation utility for foundation models over event stream data. ESGPT can represent diverse datasets across various sources in a unified manner, preprocess very large datasets extremely quickly through the use of the Polars library [33], and can tune hyperparameters, pre-train, fine-tune, and perform zero-shot evaluation for foundation models through a Hugging Face compatible API. In the rest of this work, we will demonstrate the unique value ESGPT offers through the lens of a real-world working example: building foundation models over the MIMIC-IV electronic health record dataset [15] (Section 2). We will walk through all aspects of the pipeline one would follow to pursue this research, revealing in each how ESGPT fills a critical gap in existing tools across data preprocessing (Section 3), model configuration and running (Section 4), and evaluation (Section 5). Finally, we will close with related work and concluding thoughts.

## 2 Working Example Problem Setup

**Source Data**  The MIMIC-IV dataset [15] is a publicly available dataset consisting of EHR data for all adult patients who were admitted to the emergency department or an intensive care unit (ICU) at Beth Israel Deaconess Medical Center between 2008 and 2019. This data set contains approximately 300,000 patients and consists of numerous health data modalities, including diagnoses, laboratory test results, medications, hospital and outpatient mortality, and many other variables, all localized continuously in time. For our running example, we will define the internal covariates of each event to include the set of modeling targets in Table 1.

**Modeling Task**  Our sample modeling challenge is to build a GPT model on the complex event stream data contained in MIMIC-IV. This can also be seen as a multivariate marked temporal point process. In particular, given a sequence of complex events $x_1, \ldots, x_N$ (where each event $x_i$ is a collection of occurrences of the selected covariates in Table 1) that occur at continuous times $t_1, \ldots, t_N$, we wish to model the following probability distribution:

$$p(t_i, \boldsymbol{x}_i | \underbrace{(t_1, \boldsymbol{x}_1), \ldots, (t_{i-1}, \boldsymbol{x}_{i-1})}_{\boldsymbol{h}_{i-1}})$$

We will realize this through a transformer neural network architecture parametrized by $\boldsymbol{\theta}$, such that $f_{\boldsymbol{\theta}}(t_i, \boldsymbol{x}_i, \boldsymbol{h}_{i-1}) = p(t_i, \boldsymbol{x}_i | \boldsymbol{h}_{i-1})$. Note that, unlike other GPT modeling settings, it may be the case that internal covariates of each event $\boldsymbol{x}_i$ have internal causal dependencies. For example, if $\boldsymbol{x}_i^{(j)}$ is used to denote the $j$th internal covariate of event $i$, then $p(\boldsymbol{x}_i | \boldsymbol{h}_{i-1}, t_i) \neq \prod_j p(\boldsymbol{x}_i^{(j)} | \boldsymbol{h}_{i-1}, t_i)$. Any full generative model will therefore need to account for these internal causal dependencies.

---

[1] https://eventstreamml.readthedocs.io/en/dev/index.html

| Admission/Discharge | Demographics | Measurements/Actions |
|---|---|---|
| Admission Type | Language | Laboratory Tests* |
| Admission Location | Race | Procedures |
| ICU Care-unit | Marital Status | Medications |
| Discharge Destination | Insurance Type | Infusions* |
| | Age | Diagnoses |
| | Time-of-day | Patient Weight |
| | Gender | |

Table 1: The modalities of data we include in our MIMIC-IV working example. Our generative model consists of continuous timeseries of partial observations of any of these events, and we task the model with predicting which of the times of these events, which modalities will be observed in those events, and what values they will take on. Included modalities are not observed uniformly across all events, in a non-random manner. * Includes continuous regression components.

## 3 Data Extraction and Pre-processing

### 3.1 The Problem

To build a dataset suitable for performing our task outlined in Section 2, we do the following:

1. Extract a filtered cohort of raw data from its source (here, the MIMIC-IV postgresql database).
2. Pre-process the data: learn outlier detection and normalization parameters for numerical variables, filter out infrequently measured variables or outliers, normalize data, etc.
3. Join independent input data sets together in a format optimized for deep learning.
4. Build a PyTorch dataset, dataloader, and embedding layer for this deep-learning-friendly structure so that we can use these data efficiently in downstream pipelines.

### 3.2 Current State-of-the-art

If we wish to use current solutions for data extraction, we must find an existing tool that is either fully specialized to our dataset, MIMIC-IV, and use case (of which no such example exists), or is a general purpose data extraction, preprocessing, and/or modeling pipeline that will ease our workflow. For such a general-purpose tool to be useful, it must meet several critical criteria:

1. It must solve at least a majority of the challenges identified in Section 3.1.
2. It must be flexible enough to handle not only the MIMIC-IV data, but also various other input datasets, schemas, and covariate choices.
3. It must be designed with efficiency in mind – both in data pre-processing and in the output deep-learning representations it produces, so that we can pre-train large models effectively.
4. It must be actively maintained, well documented, and tested.

Unfortunately, as shown in Table 2, existing pipelines for extracting EHR data for machine learning do not meet these criteria fully. Only one pipeline meets even a meaningful subset of these critical requirements, TemporAI [28], as it has strong development practices, exposes a flexible input format, and meets our critical pre-processing feature goals. However, TemporAI does not support native extraction from underlying source files, and suffers in its efficiency, both in the core data preprocessing pipeline and in its output deep learning representation, raising concerns about its viability on larger datasets.

### 3.3 Contributions of Event Stream GPT

ESGPT solves all these challenges in data extraction and pre-processing. Figure 1 shows the entire pipeline in visual form. From a flexible, concise, and user-friendly configuration file, ESGPT can extract raw data from a source, preprocess categorical and numerical data elements, and produce efficient PyTorch datasets for downstream analysis. In the following, we detail three key advantages of the ESGPT system. Full technical details of the pipeline are given in the Supplementary Information (SI).

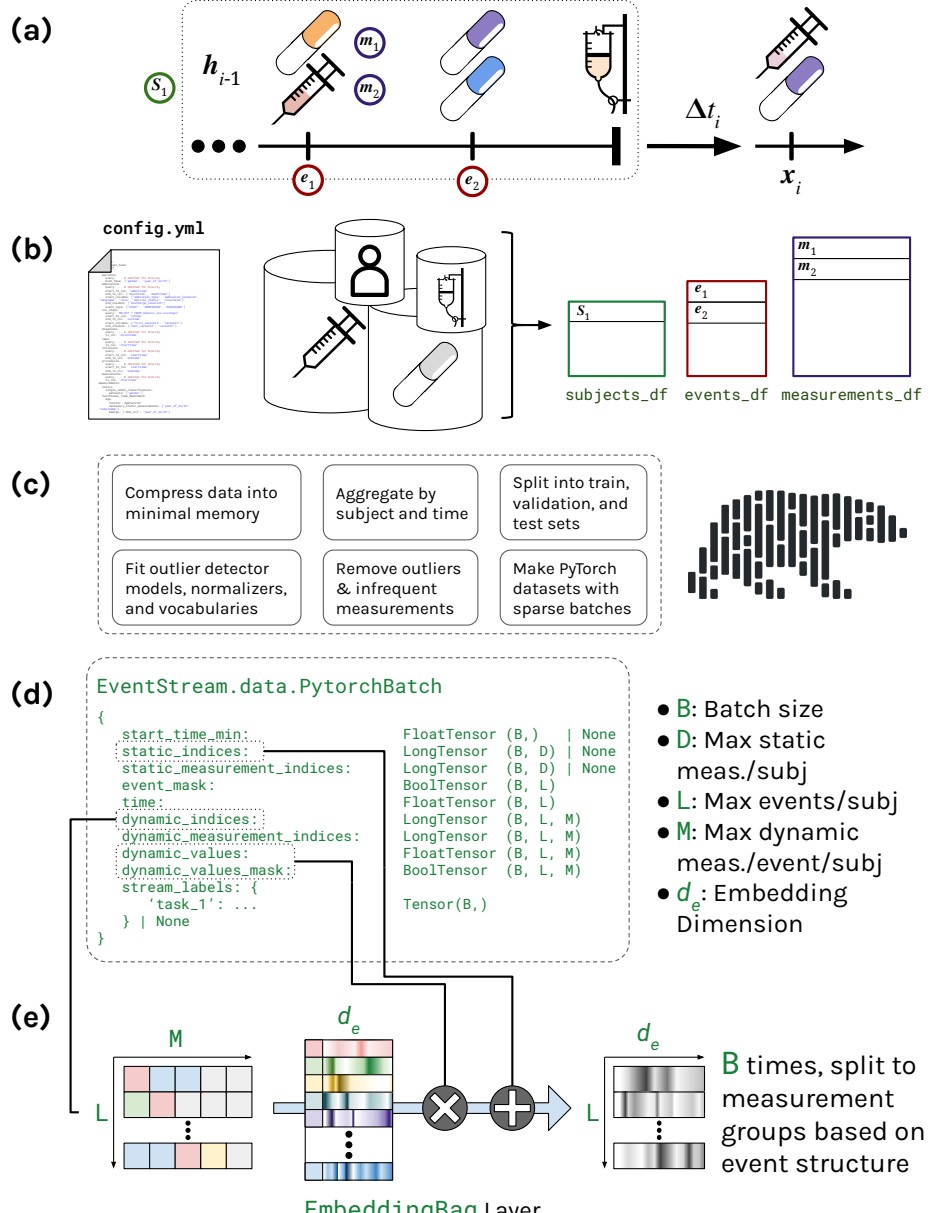

Figure 1: ESGPT's end-to-end data pipeline, which spans raw data extraction from source all the way to production of a PyTorch dataset and pre-built embedder suitable for use in any deep learning pipeline. **(a)** An example of our data modality; for a single subject $S_1$, the data consist of a series of events at continuous timestamps $e_1, e_2, \ldots$ (such as medical visits), with each event being composed of interrelated internal covariate measurements $m_1, m_2, \ldots$ (such as laboratory test results, medication prescriptions, infusions, etc.). **(b)** These data elements can be distributed among many input data sources in raw form. From a simple YAML configuration file, ESGPT can extract these data elements from the source and compile them into an internal data model consisting of three key dataframes: `subjects_df` for static data, `events_df` containing event timestamps and types per subject, and `dynamic_measurements_df`, storing all observed measurements. **(c)** ESGPT preprocesses these data frames across several critical axes, doing so efficiently through the use of the Polars library [33]. **(d)** ESGPT produces a PyTorch data set that produces batches whose size scales with the number of data elements observed per event, not with the size of the input vocabulary. **(e)** ESGPT provides a default embedding layer capable of embedding these sparse batches efficiently.

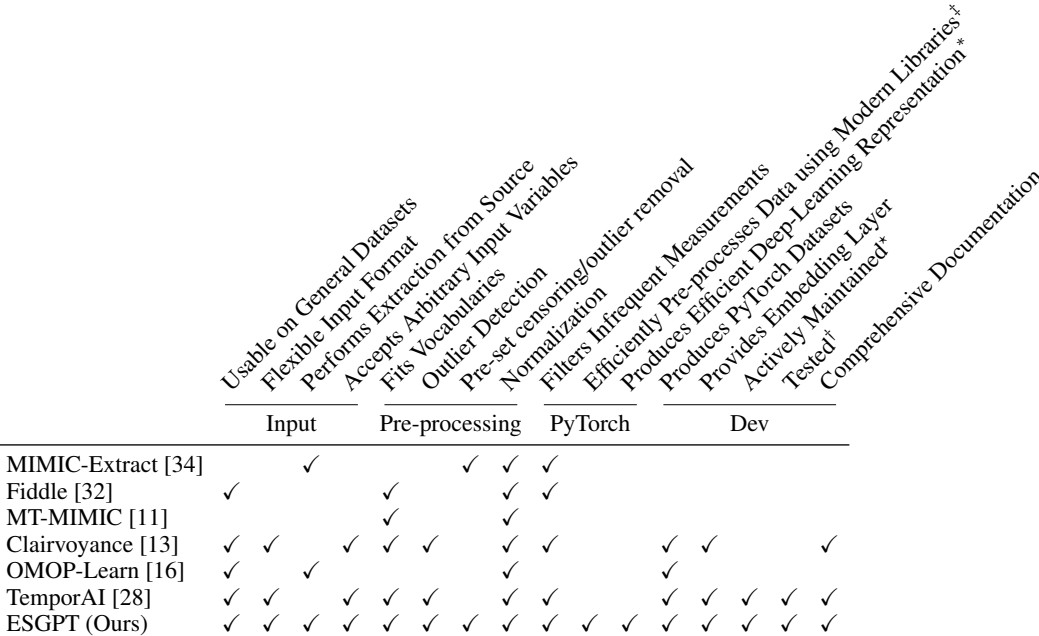

| | Input | | | | | Pre-processing | | | | | PyTorch | | | Dev | | |
| Method | Usable on General Datasets | Flexible Input Format | Performs Extraction from Source | Accepts Arbitrary Input Variables | Fits Vocabularies | Outlier Detection | Pre-set censoring/outlier removal | Normalization | Filters Infrequent Measurements | Efficiently Pre-processes Data using Modern Libraries‡ | Produces Efficient Deep-Learning Representation* | Produces PyTorch Datasets | Provides Embedding Layer | Actively Maintained* | Tested† | Comprehensive Documentation |
| --- | --- | --- | --- | --- | --- | --- | --- | --- | --- | --- | --- | --- | --- | --- | --- | --- |
| MIMIC-Extract [34] | | ✓ | | | | ✓ | ✓ | ✓ | | | | | | | | |
| Fiddle [32] | ✓ | | | | | ✓ | | ✓ | ✓ | | | | | | | |
| MT-MIMIC [11] | | | | | | | ✓ | ✓ | | | | | | | | |
| Clairvoyance [13] | ✓ | ✓ | | ✓ | ✓ | ✓ | | ✓ | ✓ | | | ✓ | ✓ | | | ✓ |
| OMOP-Learn [16] | ✓ | | ✓ | | | | | ✓ | | | ✓ | | | | | |
| TemporAI [28] | ✓ | ✓ | | ✓ | ✓ | ✓ | | ✓ | ✓ | | | ✓ | ✓ | ✓ | ✓ | ✓ |
| ESGPT (Ours) | ✓ | ✓ | ✓ | ✓ | ✓ | ✓ | ✓ | ✓ | ✓ | ✓ | ✓ | ✓ | ✓ | ✓ | ✓ | ✓ |

Table 2: A collection of existing data pre-processing and/or modeling pipelines for temporal EHR data, categorized on whether or not they enable various features. ⋆ Active maintenance is defined by the existence of a commit between May 2022 and May 2023. † Tested is defined by the presence of a reasonably comprehensive and automated test suite. ‡ Efficient data preprocessing is defined by whether or not the library relies on Pandas [35, 25] for its dataframe manipulation, as opposed to more modern, faster systems such as Polars [33] or DuckDB [22]. ∗ Efficient deep-learning representation is defined by the extent to which sparsity is leveraged in their deep-learning representation files; efficient outputs should only scale in size with the observations actually present in any given batch of data, not with the number of total features in the space, total possible sequence elements, etc.

**Event Stream GPT is easy to use and flexible**    To extract a machine learning ready data set from *any* temporal structured data set with ESGPT, a user only needs to specify a modest configuration file detailing the input data sources, the desired modalities to extract, and preprocessing pipeline parameters. This configuration file instructs the pipeline how to extract the initial data from its raw source, preprocess the extracted measurements, aggregate the data temporally, and produce deep-learning efficient representation outputs suitable for downstream modeling. A full configuration file for our working example over MIMIC-IV can be found in the SI.

**Event Stream GPT can efficiently pre-process large datasets**    Through the use of the Polars library [33] and careful design choices, ESGPT is able to provide an extremely efficient and fast pre-processing pipeline. For our working example, our entire extraction and preprocessing pipeline over the MIMIC-IV dataset (resulting in a cohort of approximately 12 thousand patients) takes only approximately 30 minutes. We further assess the performance of this pipeline on two other datasets on different compute environments, with cohort sizes of 116 thousand and 145 thousand patients, and find runtimes of 25 minutes and 62 minutes, respectively, further demonstrating the strong performance of this pipeline. Output dataset sizes are also universally small, with no dataset exceeding 4GB of final space on disk, with only standard compression. Unfortunately, there are no direct competitors that replicate all the steps of ESGPT in a consistent manner against which we can compare these results. However, to contextualize them, in order to run omop-learn on MIMIC-IV, one would first need to convert MIMIC-IV to the OMOP format, a process that is supported by existing ETLs. Running *only* these existing ETLs requires a runtime of over 37 minutes,[2] meaning that *the entire ESGPT data preprocessing pipeline, which runs from scratch on MIMIC-IV and includes*

---

[2] https://github.com/OHDSI/MIMIC/blob/df97a75cd974c491e595c8b007a79f7326066cb1/z_more/run_times.txt#L4

| Pipeline | Disk (GB) | RAM (GB) | Load Time (sec) | Time/Batch (sec) | GPU Memory (MB) |
|---|---|---|---|---|---|
| TemporAI | 3.0 | 413.5 | 229.5 | $0.6 \pm 0.0$ | $877.2 \pm 95.5$ |
| omop-learn | 4.2 | 17.2 | 133.9 | $\mathbf{0.4 \pm 0.0}$ | $\mathbf{40.1 \pm 19.7}$ |
| ESGPT (ours) | **0.6** | **7.6** | **42.5** | $\mathbf{0.4 \pm 0.0}$ | $67.2 \pm 35.0$ |

Table 3: Performance comparisons of the output deep-learning datasets between TemporAI [28], omop-learn [16], and ESGPT (our pipeline) over the MIMIC-IV example. We can see that ESGPT is dramatically cheaper than both other pipeline in terms of full dataset storage cost, and is cheaper or competitive with both other pipelines on batch iteration time, and is slightly more expensive than omop-learn in terms of batch memory but dramatically cheaper than TemporAI.

*extraction, preprocessing, and final representation, all happens faster than just the initial extraction and data conversion stage required for omop-learn.* Full details are provided in the SI.

**Event Stream GPT provides a memory-efficient deep learning representation** Beyond efficient preprocessing of data, ESGPT also provides an efficient deep learning representation of the data suitable for PyTorch modeling. In particular, ESGPT provides a data representation format that is simultaneously easy to use, through provided PyTorch dataset, batch, and embedding layer objects; efficient to load and work with from disk, in terms of both time and memory; and permits rapid and memory efficient construction of PyTorch batches for use in deep learning models on the GPU. To demonstrate the impact of these properties, we compare ESGPT to TemporAI and omop-learn (two of the most comparable existing pipelines from Table 2) on metrics quantifying the time and memory costs of working with deep-learning datasets through each of the three pipelines in Table 3 for our working MIMIC-IV example. We can see from these comparisons that ESGPT is dramatically more efficient in terms of time, storage, and memory cost to work with these data than either other pipeline, and is competitive with omop-learn and dramatically better than TemporAI in terms of dataset iteration time and GPU memory cost. Notably, we must recognize that as neither TemporAI nor omop-learn are directly comparable to ESGPT, these comparisons are approximate, may not reflect the ideal use-cases of the respective pipelines, and required us to craft new code to approximate the common way in which users would construct a PyTorch dataset when using TemporAI or omop-learn; nevertheless, this demonstrates that for its intended use-case, ESGPT provides significant advancements over existing options in terms of computational performance when working with large scale datasets. We comment further on these comparisons in the Supplementary Information, in particular providing more details on the assumptions and approximations required.

## 4 Building GPTs

### 4.1 The Problem

To produce the generative model outlined in Section 2, we must satisfy several desiderata: (1) We need to be able to model historical dependencies ($p(t_i, \boldsymbol{x}_i | \boldsymbol{h}_{i-1})$), temporal dependencies ($p(\boldsymbol{x}_i | t_i)$), and intraevent dependencies ($p(\boldsymbol{x}_i^{(j+1)} | \boldsymbol{x}_i^{(1)}, \ldots, \boldsymbol{x}_i^{(j-1)})$). In contrast, traditional (sequential) GPTs only need to model historical dependencies. (2) We need to enable deep, sequential processing of event sequences that can take into account event timestamps to build high-capacity representations. (3) We need to produce continuous, categorical, and temporal output emission distributions. In contrast, traditional GPTs only need to output categorical emission distributions. (4) Our model must be able to construct time-derived features (*e.g.*, age) on the fly during generation.

### 4.2 Current State-of-the-art

There are three categories of existing tools that could help us build models: General-purpose deep learning modeling frameworks with built-in model architectures (*e.g.*, TemporAI [28]); other GPT architectures for different problem settings, such as natural language or regularly sampled timeseries; or existing models for autoregressive or generative modeling of EHR or related modalities.

Unfortunately, across all three options, there are no existing tools that satisfy the criteria outlined in Section 4.1. Among existing general-purpose frameworks, such as TemporAI [28] or OMOP

| Model | TTE | Continuous Outputs | Categorical Outputs | Intra-event Dependencies | Generative | Fine-tuning | Zero-shot |
|---|---|---|---|---|---|---|---|
| | | Models | | | Evaluation | | |
| Doctor AI [3] | ✓ | | ✓ | | ✓ | | |
| CDT [18] | | | ✓ | | | | ✓ |
| MetaCare++ [31] | ✓ | | ✓ | | ✓ | | |
| CLMBR [30] | | | ✓ | | | ✓ | |
| MedTPP [6] | ✓ | | ✓ | | ✓ | | |
| MedGPT [17] | ✓ | | ✓ | | ✓ | | |
| **ESGPT** | ✓ | ✓ | ✓ | ✓ | ✓ | ✓ | ✓ |

Table 4: A summary of existing foundation models over EHR data, broken down by what modalities of data they model and how they report evaluation results.

Learn [16], they lack any suitable built-in model architectures for this task. Existing GPT architectures, on the other hand, can be very helpful, especially for providing sources to adapt transformer architectures for core multilayer network structures, but lack key features specific to this modality, such as accounting for unknown output event times and complex interevent dependencies. Finally, among existing, published generative models for EHR or other similar modalities, all existing architectures have major limitations, as outlined in Table 4.

### 4.3 Contributions of Event Stream GPT

The ESGPT library includes a pre-built model architecture that models time-to-event components, continuous and categorical output measurements, handles intra-event dependencies in a user-configurable way, and provides a HuggingFace-compatible API for generating new events with a pre-trained model. We detail these capabilities below, and a full technical reference is in the SI.

**Event Stream GPT provides a HuggingFace API compatible interface for building models capable of outputting complex, temporal distributions**   To use one of the pre-defined model architectures in ESGPT, users simply build a specialized configuration object, which is a sub-class of the standard, Hugging Face `PretrainedConfig`[3]. Once the config is defined, a model can be instantiated in the same way as any other Hugging Face pre-trained model, either from scratch or from a pre-trained source. In addition to the standard Hugging Face attributes, the ESGPT config also contains several custom attributes specific to our setting, including attributes describing the breakdown of the various measurements and their respective vocabulary sizes, the implied dependency relationships between intra-event covariates, and data embedding configuration parameters.

All pre-built ESGPT models for generative modeling output a unified model output object. This object maps event generation covariate target names to PyTorch distributions, from which we can easily compute losses or sample new events. The unified ESGPT dataset class and this output object defines the API expected by ESGPT models in further utilities like generative capabilities.

**Event Stream GPT provides a modeling interface for encoding complex, custom intra-event dependencies**   ESGPT comes with two pre-defined model architectures. One (the `ConditionallyIndependent` model) that matches much of the existing published literature and only supports defining generative models where all intra-event covariates are conditionally independent from one another given the patient's history and one (the `NestedAttention` model) that supports user-defined dependency relationships between various internal features of $x_i$. Users specify these relationships through the configuration file. For example, one of the models pre-trained in our MIMIC-IV working example relies on the intra-event dependency chain shown in Figure 2. This specification takes the form of a list of lists, and tells the model that it should make the assumption

---

[3]https://huggingface.co/docs/transformers/main_classes/configuration

```
structured_event_processing_mode: nested_attention
measurements_per_dep_graph_level:
    - ["age", "time_of_day"] # Must start with FUNCTIONAL_TIME_DEPENDENT measurements.
    - [
        "event_type", "patientweight", "admission_type", "admission_location",
        "race", "language", "marital_status", "insurance", "careunit",
        ["lab_itemid", "categorical_only"], ["infusion_itemid", "categorical_only"]
      ]
    - [["lab_itemid", "numerical_only"], ["infusion_itemid", "numerical_only"]]
    - ["procedure_itemid", "medication", "icd_code", "discharge_location"]
```

Figure 2: A sample dependency graph for our MIMIC-IV working example. All measurements within any inner list depend only on those measurements in prior lists.

that the internal covariates in a given inner list of $x_i$ are conditionally independent of one another given both (1) the historical representation of all $t_\ell, x_\ell, \ell < i$ and (2) the internal covariates in all prior lists in the specified dependency graph. Concretely, this means that the model configuration snippet shown in Figure 2 defines a generative model such that the predicted type of an event will depend only on the historical representation and the true time of that event, but the predicted values observed for the laboratory tests ordered for this patient will depend on both the time of the event, but also the event type, patient dynamic demographics, and the categorical laboratory test item IDs that are actually ordered for that patient.

We describe the architecture used to model this in full detail in the SI; however, what is more essential for the utility of the ESGPT software library is that this API for specifying internal dependencies, and the associated non-architectural changes required to support it (*e.g.*, enabling generation to respect the multi-stage process implied by the intra-event dependency graph) can be used across a variety of internal transformer architectures. Thus, this provided model class can serve as a basis for further model development that relies on internal event dependencies in future research without invalidating the benefits of the other tools in ESGPT. While the `NestedAttention` model builds on pre-existing works, to the best of our knowledge this precise architecture is novel.

**Event Stream GPT provides generation capabilities for unsupervised, zero-shot evaluation**   A critical use-case for foundation models is the ability to generate new, unseen predictions for the future of an input. To the best of our knowledge, ESGPT is the first pipeline to bring that capability in a systemized fashion to foundation models over event stream data like EHR systems. To do so, we introduce several key modifications from the traditional Hugging Face generation API, including the ability to support complex emission distributions during generation, dynamic computation of time-dependent input features for newly generated time-points, and generating events in sequence while respecting the causal path. Full details can be found in the SI.

## 5   Evaluating Foundation Models

### 5.1   The Problem

Even after building a foundation model on this non-standard domain, one still has to evaluate that model, both as a concrete representation learning system and in its possible viability as an early stage foundation model. In domains of complex, timeseries of events, like medical record datasets, there are a number of unique challenges that must be addressed in evaluation. We foresee the following metrics to be critical points to assess in early stage foundation models in such domains: (1) Performance as a raw generative model, (2) Viability as a foundation model, (3) Practical utility vs. current ML systems, (4) Performance disparities across subject sub-populations, and (5) Privacy risks.

### 5.2   Current State-of-the-art

Existing models in this space have significant gaps in their evaluation strategies to date. Table 4, in addition to summarizing model modalities, also summarizes evaluation methods used. While some techniques like few-shot fine-tuning analyses are well represented, others are under-explored. *In*

*particular, at present there are no existing strategies to reliably assess general, zero-shot performance of GPTs over event stream datasets, a gap which we fill here.*

## 5.3 Contributions of Event Stream GPT

ESGPT focuses on providing usable, powerful tools to enable rapid evaluation of GPTs over event stream datasets, focused currently on the assessment of a model's generative performance and in its viability as a foundation model in few and zero-shot evaluation tasks. While further efforts on the other evaluation criteria would be highly impactful, given the dearth of existing utilities just on assessing these basic performance metrics, we leave the rest to future work. In addition to these evaluation utilities, we also provide pre-built scripts for running distributed Bayesian hyperparameter optimization sweeps over foundation models through Weights and Biases (WandB) [2].

**Event Stream GPT assesses generative model performance & hyperparameter tuning**    To make assessing generative performance easier, ESGPT comes pre-equipped with a PyTorch Lightning [7] module for running pre-training models which tracks a variety of metrics to assess generative model performance, including traditional classification and regression metrics over all component internal event covariates. These are logged by default to WandB, permitting easy exploration of output model performance. These metrics are further tracked and can be assessed visually in WandB reports during hyperparameter tuning as well, which is run via a WandB sweep with a provided default hyperparameter search space. A template report for assessing the output of these sweeps is also provided alongside the code for our working example over MIMIC-IV (accessible in the SI), allowing users to easily design their own dashboards to monitor their model tuning runs.

**Event Stream GPT assesses foundation model viability via few and zero-shot performance**
While many systems already explore few-shot performance (and ESGPT comes pre-equipped with lightning modules and utilities for performing similar assessments), assessing zero-shot performance across general models has not been previously studied. Enabling zero-shot evaluation for foundation models trained on event stream data requires three things: First, a labeled dataset corresponding to a well-defined task which can be used to evaluate zero-shot produced labels; second, a method to perform unsupervised forecasting of a patient's record given an input window, which ESGPT provides; third, a function to translate generated samples into empirical predictions.

The ESGPT system then provides a PyTorch Lightning utility to take as input a dataset, task, pre-trained model, and labeling function, and perform zero-shot evaluation over a specified data cohort. This system reports performance metrics based on the output of this labeling function over the empirical predictions generated for each patient's input. *This provides any model trained using the ESGPT API the ability to assess zero-shot performance in a meaningful way, out of the box.* For examples of these evaluation utilities on MIMIC-IV, see the SI.

## 5.4 Example ESGPT Models over MIMIC-IV

To demonstrate the evaluation capabilities of ESGPT, in this section we provide examples of evaluating pre-trained models or training supervised models from scratch on two canonical fine-tuning tasks over MIMIC-IV: In hospital mortality prediction and 30-day readmission prediction (though all of these capabilities are fully extensible to any binary classification task over an ESGPT dataset). Full details of these experiments and results can be found in the online documentation for the MIMIC-IV tutorial[4], but here we highlight three capabilities of ESGPT: (1) producing baseline performance numbers for fine-tuning tasks over ESGPT systems, (2) performing traditional fine-tuning, and (3) performing zero-shot evaluation of a pre-trained model.

In particular, with regard to baselines, Event Stream GPT provides utilities to hyperparameter tune baselines across both the same neural network architectures as are used during generative pre-training and scikit-learn [26] classifiers. On our proof-of-concept MIMIC-IV cohort for the in-hospital mortality prediction task, for example, hyperparameter tuned random forest classification pipelines can obtain a held-out AUROC of approximately 0.67, and for the readmission risk prediction task, it obtains a performance of approximately 0.65. On the other hand, training neural network architectures from scratch yields AUROCs of 0.79 and 0.57 for the two tasks, respectively.

---

[4]`eventstreamml.readthedocs.io/en/dev/MIMIC_IV_tutorial/index.html`

When fine-tuning a model from a pre-trained model, we support limited hyperparameter tuning over just non-fixed parameters (e.g., learning rate, batch size, etc.). Doing this for the in-hospital mortality and readmission risk prediction tasks achieves AUROCs of approximately 0.8 and 0.62, respectively, in both cases yielding a lift over from-scratch training due to the generative pre-training at the 12k patient dataset scale. We can also assess performance via zero-shot performance. While our proof of concept models do not show better than chance AUROC on these measures, it is nonetheless a significant strength of ESGPT that these analyses can now be performed in a systematic manner.

## 6 Further Discussion

**Limitations** While ESGPT provides significant benefits, it also has a number of limitations. Firstly, it lacks some more advanced pre-processing capabilities that other systems support, such as unit conversion, ontological aggregation, or support for a wide variety of optional pre-processing algorithms. Secondly, defining downstream tasks is currently only possible through manual creation / user-defined code, as opposed to via a configuration language. Next, while the utilities ESGPT offers do significantly enhance evaluation, even more is needed on that front, including dedicated metrics for fairness and privacy sensitivity, methods to assess embedding space structure, and methods to detect emergence of foundation model capabilities more cost efficiently. As a mitigation for these issues, we recommend that current users of these pipelines rigorously assess model performance for any downstream tasks across relevant patient subgroups to ensure bias or fairness implications are appropriately measured. Further research on this point, extending upon efforts such as those explored in [5, 37] is needed. Privacy mitigation strategies also need significantly more exploration before models can be safely released. While prior work on NLP models have found some risks due to memorization, the literature is far from clear on the risks or ways to address them for these kinds of models [12, 19]. Relatedly, any downstream users of ESGPT who wish to release datasets pre-processed with ESGPT should also ensure they take all appropriate precautions to ensure ethical, safe dataset release, and include sufficient documentation of the dataset's provenance, composition, and viable use-cases or pit-falls to ensure effective, impactful downstream use [9]. Thirdly, ESGPT currently only supports categorical or continuous valued event stream data modalities; however, this pipeline could also be extended to support additional modalities such as wearable and signals data, with similar abilities as those described in [21, 8]. Finally, ESGPT provides primary value as a data pre-processing pipeline and set of modeling APIs, not as a static set of model architectures or pre-trained weights. As such, we leave detailed comparisons of different model architectures that can be built using the APIs provided by ESGPT across a variety of benchmark tasks to future work.

**Related Work** As listed in Tables 2 & 4, there are a number of existing processing pipelines and models of relevance here. Beyond those already discussed, there are also a number of models that leverage nongenerative losses that warrant highlighting, such as contrastive methods like [1, 20, 23]. These provide a valuable alternative to generative foundation models in various domains. Additionally, other pipelines also exist in this and adjacent spaces, such as time series specific pipelines and tabular data embedding methodologies [14, 10, 4]. GPTs have also been used very successfully in other domains outside of event stream data. These include both examples of foundation models outside of either NLP or event stream data, such as on protein sequences [27] or point clouds [36], but also other examples of temporal point processes outside of the typical "foundation model" framework [29, 38].

## 7 Conclusion

In this paper, we introduced Event Stream GPT (ESGPT), a new, open source software library that fills a critical gap in the foundation model literature: namely, technological support for building generative, pre-trained transformers (GPTs) over "event stream data"—datasets consisting of continuous time sequences of complex events with multiple modalities and internal dependencies. We showed that Event Stream GPT can reliably address all aspects of the research pipeline for this end, from data pre-processing at scale, model construction in a manner accounting for the unique properties of event stream data, and evaluation across all aspects of the foundation model pipeline. We believe that ESGPT will significantly accelerate, standardize, and improve the space of research into these models and will help identify the best architectures and settings where the strengths of GPTs as seen in other domains can be replicated in new domains.

## Acknowledgments and Disclosure of Funding

MBAM is supported by a Berkowitz Postdoctoral Fellowship at Harvard Medical School.
BN completed this work while in the Department of Computer Science at the University of Toronto.
This work has benefited from the advice of Dr. Oleksander Shchur.
The All of Us Research Program is supported by the National Institutes of Health, Office of the Director: Regional Medical Centers: 1 OT2 OD026549; 1 OT2 OD026554; 1 OT2 OD026557; 1 OT2 OD026556; 1 OT2 OD026550; 1 OT2 OD 026552; 1 OT2 OD026553; 1 OT2 OD026548; 1 OT2 OD026551; 1 OT2 OD026555; IAA #: AOD 16037; Federally Qualified Health Centers: HHSN 263201600085U; Data and Research Center: 5 U2C OD023196; Biobank: 1 U24 OD023121; The Participant Center: U24 OD023176; Participant Technology Systems Center: 1 U24 OD023163; Communications and Engagement: 3 OT2 OD023205; 3 OT2 OD023206; and Community Partners: 1 OT2 OD025277; 3 OT2 OD025315; 1 OT2 OD025337; 1 OT2 OD025276. In addition, the All of Us Research Program would not be possible without the partnership of its participants.

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
