# Supplementary Information

## 1  Code and Full Technical Documentation of Event Stream GPT

Event Stream GPT is open source, available at `https://github.com/mmcdermott/EventStreamGPT`. Extensive online documentation is accessible at `https://eventstreamml.readthedocs.io`. This has full API details, usage guide, and descriptions of all aspects of the pipeline, all in an easy to read format.

**Full Technical Documentation**  Full technical documentation for ESGPT can be found in online at `https://eventstreamml.readthedocs.io`. It features:

1. API: A full API reference of all module classes, functions, configuration files, etc.
2. MIMIC-IV Tutorial: A full walk-through of our MIMIC-IV example, both in written documentation and in code, with concrete examples of model and evaluation output results over MIMIC-IV, as well as a link to a stand-alone github repository for that example.
3. Dataset Pipeline: A full description of the dataset pipeline and configuration language.
4. Pre-built Architectures: Full descriptions of our two pre-built model architectures.

## 2  Computational Performance on MIMIC-IV and Additional Datasets

The ESGPT pipeline is not specific to MIMIC-IV. We have used it on a number of other datasets, spanning both public and private settings. In Table 1, we show a summary of the pipeline's usability across three different datasets: the public MIMIC-IV dataset [6], the public AllOfUs Dataset [5], which is a large, electronic health record (EHR) dataset consisting of broad spectrum care over a large number of individuals from diverse areas, health statuses, and backgrounds; and the "HF" dataset, a private dataset sourced from an ongoing, IRB-approved study regarding the use of machine learning for analysis of heart failure measurements from an academic research institution consisting of labs, vitals, and cardiology derived measures for a large number of patients. We see that the system is able to process large datasets very quickly with only modest configuration inputs from the user with consistent space reductions in all three settings.

## 3  Performance comparisons against TemporAI and omop-learn

In the main body Table 2, we compare against a variety of existing pipelines for data pre-processing and modeling over EHR style data. We further quantify these comparisons in the main body, Table 3, focusing specifically on aspects of "deep learning dataset" creation and usage. In this section, we expand upon these comparisons and offer insights as to why the performance of the various pipelines differ so greatly. In this table, the columns refer to the following quantities: "Disk" refers to the amount of space on disk the stored files take, with standard compression, in GB; "RAM" refers to the amount of RAM required to load the full deep-learning dataset into memory; "Load Time" measures the time required to load the full deep learning dataset into memory; "Time/Batch" refers to the amount of time required to iterate through a single batch of size 16, presented as average plus or minus standard deviation.; "GPU Memory" refers to the amount of GPU memory that would be required to store the data in said single batch of size 16, presented as average plus or minus standard deviation. Lower is better on all metrics. Note that TemporAI and omop-learn numbers are all approximate, as these pipelines do not have a fully unified deep learning iteration data format /

37th Conference on Neural Information Processing Systems (NeurIPS 2023).

| Dataset | Public? | Input Format | Input Size | |
|---|---|---|---|---|
| | | | # Subjects | Disk (GB) |
| MIMIC-IV [6] | ✓ | PostgreSQL | 300k | 43 |
| AllOfUs [5] | ✓ | Parquet | 159k | 11 |
| HF | | CSV | 134k | 32 |

| Dataset | # Lines | CPUs | Time (min) | Mem (GB) | Output | | | Disk (GB) | |
|---|---|---|---|---|---|---|---|---|---|
| | | | | | Subj. | Events | Meas. | Non-DL | DL |
| MIMIC-IV | 153 | 10 | 30.78 ± 1.12 | 65.93 ± 0.14 | 12k | 3M | 222M | 1.19 | 0.49 |
| AllOfUs* | 98 | 64 | 62.03 ± 0.31 | 119.36 ± 0.93 | 145k | 28M | 330M | 2.95 | 3.99 |
| HF | 78 | 128 | 24.96 ± 0.44 | 161.07 ± 0.23 | 116k | 25M | 244M | 3.6 | 1.54 |

Table 1: *Top:* Descriptions of the raw data for each of our four datasets. *Bottom:* Statistics on computational cost to produce each dataset. "# Lines" refers to the number of lines in the 'yaml' configuration file used to produce this dataset. Standard deviations are omitted when they are universally $\leq 0.0$. Memory was assessed via `mprof` for MIMIC-IV and HF, but due to technical issues it was assessed via a shell loop with the 'free' command for AllOfUs.

style, so commonly used elements were re-implemented on the same dataset to form these numbers. For both TemporAI and omop-learn, measurements were made on a random subset of patients and appropriately extrapolated within the dataset to minimize computational cost for these experiments. Additional details and commentary on both comparison points are present below. Additionally, both TemporAI and omop-learn lack many capabilities that ESGPT has, such as the inclusion of embedding layers, clear masks provided with the batches, the ability to sub-sample different sequences of the maximum permitted length per epoch, etc.

**TemporAI comparisons** TemporAI scores very poorly on these metrics because it uses a wide representation format across dataset "vocabulary" elements compared to the long format that ESGPT uses. In particular, on MIMIC-IV, TemporAI's data format requires instantiating a column for every lab test in the dataset's vocabulary. These columns are further included in the final tensorized representation of the data at a batch level, thus meaning that the batch format of TemporAI scales in memory cost with the total vocabulary size of the dataset, which is very large. Beyond this, TemporAI also permits only one recorded measurement of a given column per unique timestamp, thus meaning that a full summary must actually include multiple columns per, for example, lab test, to capture the mean, count, minimum, etc. of the set of values observed in that time bucket. In contrast, ESGPT's format is a long, sparse format, meaning the GPU memory footprint scales only with the number of observations in a patient's event, not with the overall vocabulary size. This memory cost disparity also makes it slower to load and iterate through TemporAI's dataset. Their dataloading code is, actually, simpler and likely faster than ESGPT's when compared on equal memory batches; however, because TemporAI batches contain so much additional information (which is not necessary as most of that information corresponds to missing laboratory test values and dense tensors are used), the cost on a batch of the same number of patients takes longer to load.

**omop-learn comparisons** For omop-learn specifically, the pipeline does not natively include numerical values, only codes, so to make this comparison fair it was altered to also track values in a similar manner. omop-learn's batch format is actually relatively consistent with ESGPT's (especially when modified to include values). In fact, the only reason that it performs better on memory cost here is because it does not include additional metadata that ESGPT tracks per batch, such as the type of each observation in the dataset (e.g., that a particular observation is of type laboratory test as opposed to medication) and the masks explicitly flagging missing values. If the memory cost of this additional metadata is removed, the memory cost of the two pipelines are statistically insignificantly different. However, the reason omop-learn's other metrics are worse than ESGPT (or equivalent, for time/batch), is because omop-learn's base storage format uses JSON to store the underlying measurements, which is a very memory intensive storage format due to the necessity to store string keys. This inflates the costs to load or store the dataset, and also adds additional costs to the iteration time due to the requirement to convert out of that format per patient, which likely accounts for the

| Task | # Samples | Median Samples / Patient | Prevalence |
|---|---|---|---|
| 30-day Readmission Risk | 70614 | 4 | 0.33 |
| In-hospital Mortality | 21812 | 1 | 0.09 |

Table 2: Cohort statistics for the fine-tuning tasks tested on the sample MIMIC-IV cohort.

fact that ESGPT and omop-learn are equivalent w.r.t. time per batch despite omop-learn's smaller batch memory footprint.

# 4 Proof-of-concept Experiment Details on MIMIC-IV

In the provided MIMIC tutorial[1], we show experimental results for using ESGPT to perform pre-training and fine-tuning analyses over the prediction tasks of in-hospital mortality prediction and readmission risk prediction. In this section, we provide further details on those experiments.

## 4.1 Task Details

Task details are summarized in Table 2. Both tasks are binary prediction tasks. In-hospital mortality predicts, given the first 24 hours of a patient's ICU stay admission, whether that patient will die in the hospital. Patients who die or are discharged within the first 48 hours after their ICU stay admission are excluded from the cohort. Readmission risk prediction predicts, given the patients record up until the time of discharge from the hospital, whether or not the patient will be re-admitted within 30 days.

## 4.2 Model Details

**Random Forest Pipelines**    Random forest baselines consisted of the following steps:

1. First, ESGPT data was written to a flat, historically summarized format via built-in library methods. These data summarize patient records by computing one-hot categorical counts and numerical means, variances, minimums, maximums, and counts with values across all measurements in the data over specifiable history windows.

2. Second, the baseline pipeline extracts a subset of (a) historical windows and (b) measurements to include via controllable hyperparameters.

3. These extracted features are all then scaled via a standard scaler.

4. Missing values are imputed, where the imputation strategy is chosen via hyperparameter search.

5. Next, data dimensionality are reduced via PCA, with a chosen number of output components via hyperparameter search.

6. Finally, the outcome is modeled using a random forest, again tuned via hyperparameter search.

The parameters for this hyperparameter search can be found in the source code, as can all further details about this pipeline. Hyperparameter search results for these models can be found at the below links for the two tasks:

1. Readmission risk prediction: `https://wandb.ai/mmd/MIMIC_FMs_Public/reports/30d-Readmission-Prediction-Scikit-learn-Baseline-Hyperparameter-Tuning--Vmlldzo1MjE1MjM3`

2. In-hospital mortality prediction: `https://wandb.ai/mmd/MIMIC_FMs_Public/reports/In-hospital-mortality-Scikit-learn-Baseline-Hyperparameter-Tuning--Vmlldzo1MTc5MTA4`

---

[1] `https://eventstreamml.readthedocs.io/en/dev/MIMIC_IV_tutorial/index.html#evaluating-pre-trained-models`

| Task | Random Forest AUC | From-scratch AUC | Fine-tuned AUC |
|---|---|---|---|
| Readmission risk | 0.65 | 0.57 | 0.62 |
| In-hospital Mortality | 0.67 | 0.79 | 0.80 |

Table 3: Held-out set AUROC during hyperparameter tuning across both baselines and fine-tuned models. Higher is better.

**Neural Network Baselines**   Neural network baselines were constructed with the same architectural class and hyperparameter search options as the core pre-training model architectures. The system searched over hyperparameters controlling network depth and width, style of input embeddings, nested or conditionally independent attention, regularization, and optimization parameters. Hyperparameter search results for these models can be found at the below links for the two tasks:

1. Readmission risk prediction: `https://wandb.ai/mmd/MIMIC_FMs_Public/reports/30d-Readmission-Prediction-From-scratch-Training-Hyperparameter-Tuning---Vmlldzo1MTQ4NTk4`

2. In-hospital mortality prediction: `https://wandb.ai/mmd/MIMIC_FMs_Public/reports/In-hospital-mortality-From-scratch-Training-Hyperparameter-Tuning---Vmlldzo1MjE1MzAx`

**Fine-tuned Models**   Fine-tuned models were initialized from the optimal pre-trained model selected from a pre-training hyperparameter search,[2] with the hyperparameter search limited to just those hyperparameters that could be changed without affecting network shape (so that pre-trained weights could still be used), including regularization and optimization hyperparameters. Hyperparameter search results for these models can be found at the below links for the two tasks:

1. Readmission risk prediction: `https://wandb.ai/mmd/MIMIC_FMs_Public/reports/30d-Readmission-Prediction-Fine-tuning-Hyperparameter-Tuning--Vmlldzo1MTQwNTI3`

2. In-hospital mortality prediction: `https://wandb.ai/mmd/MIMIC_FMs_Public/reports/In-hospital-mortality-Fine-tuning-Hyperparameter-Tuning--Vmlldzo1MjE1MzE2`

### 4.3   Final Results

Final results for these experiments are shown in Table 3. While the focus of this work is on the capabilities of this data-preprocessing pipeline and set of modeling APIs, not on individual model performance, we do nevertheless find that fine-tuning over a generative pre-trained model here presents modest improvements over from-scratch training, and for in-hospital mortality over a random forest baseline as well. However, there are many caveats to these experiments, including limitations to the number of hyperparameter search samples that were able to be performed, the lack of assessments of other scikit-learn style pipelines, and the lack of other feature aggregation functions in the flat representations of these data. So, these results should be taken merely as a proof of concept, not as a true demonstration of the superiority of this class of models over any other.

## 5   Further Details on Existing Models

There are a variety of existing foundation model attempts over EHR and related datasets. In Tables 4 and 5, we summarize a variety of existing models over a larger class of options than is shown in the main-body Table 3 (in particular, here we also include Contrastive and Masking based pre-training systems). Even across these broader examples, we see that evaluation is largely limited to fine-tuning evaluations, establishing a clear gap regarding zero-shot evaluation.

---

[2]`https://wandb.ai/mmd/MIMIC_FMs_Public/reports/08-10-Hyperparameter-Tuning-Sweep--Vmlldzo1MTExODQ0`

**Table 4**

| Style | Model | Primary Care | Hospital | Intensive Care Unit | Claims | Clinical Notes | Diagnoses | Procedures | Medications | Lab Orders | Lab Results | Vitals | "Token" | Patients | T/P | C/T | # Codes |
|---|---|---|---|---|---|---|---|---|---|---|---|---|---|---|---|---|---|
| | | Care Modalities | | | | Event Types | | | | | | | | | | | |
| Causal | Doctor AI [2] | ✓ | | | | ✓ | ✓ | ✓ | | | | | Visit | 0.3M | 54.6 | 3.2 | 1.8k |
| | CDT [9] | | ✓ | | | | ✓ | ✓ | ✓ | | ✓ | | Event | 0.0M | 13.6 | | 0.0k |
| | MetaCare++ [22] | | | ✓ | | | ✓ | | | | | | Visit | 0.0M | 2.2 | 4.8 | 0.9k |
| | CLMBR [21] | | ✓ | | | ✓ | ✓ | ✓ | ✓ | | | | 1 day | 3.4M | 7.0 | 5.0 | 21.7k |
| | MedTPP [3] | | | ✓ | | | ✓ | | | | | | Event | 0.0M | 4.0 | | 0.1k |
| | MedGPT [7] | | | | ✓ | ✓ | | | | | | | Event | 0.6M | | | |
| Contrastive | OCP [1] | | | | ✓ | | | | | | | | Event | 0.1M | 12.0 | | 30.5k |
| | Hi-BEHRT [10] | ✓ | ✓ | | | ✓ | ✓ | ✓ | ✓ | | | | Visit | 2.8M | 62.0 | 4.5 | 3.7k |
| Masking | Med-BERT [16] | ✓ | ✓ | | | | ✓ | | | | | | Visit | 28.5M | | | 82.0k |
| | BEHRT [11] | ✓ | ✓ | | | | ✓ | | | | | | Visit | 1.6M | | | 0.3k |
| | Graph-Transformer [14] | | | ✓ | | | | ✓ | ✓ | ✓ | ✓ | | 1 hr | 0.0M | 24.0 | | |
| | EHR-PT [12] | | | ✓ | | | | ✓ | ✓ | ✓ | ✓ | | 1 hr | 0.1M | 70.6 | | |
| | RAPT [18] | | | ✓ | | | | | ✓ | ✓ | ✓ | | Visit | 0.1M | 6.8 | | |
| | CEHR-BERT [13] | | | ✓ | | | ✓ | ✓ | ✓ | ✓ | | | Visit | 2.4M | 14.0 | 5.4 | |
| | GRACE [17] | | | ✓ | | | | | ✓ | ✓ | ✓ | | Visit | 0.0M | 6.5 | | |
| | CMS LDS BERT [8] | | | | ✓ | | ✓ | ✓ | | | | | Event | 1.2M | | | 20.0k |
| | MTL GPR [20] | | | | ✓ | | | | | | | | 1 hr | 0.0M | 38.0 | | |
| | T-BEHRT [15] | ✓ | ✓ | | | | ✓ | | | | | | Visit | 6.8M | | | |

Table 4: A summary of existing MedLMs. *T/P* refers to "tokens" per patient, *C/T* to codes per "token," and *# Codes* to the number of unique codes present in the dataset. Missing values reflect quantities not reported in the source publication.

**Table 5**

| Style | Model | TTE | Continuous Outputs | Categorical Outputs | Intra-event Dependencies | Generative | Fine-tuning | Zero-shot |
|---|---|---|---|---|---|---|---|---|
| | | Models | | | | Evaluation | | |
| Causal | Doctor AI [2] | ✓ | | ✓ | | ✓ | | |
| | CDT [9] | | | ✓ | | | | ✓ |
| | MetaCare++ [22] | ✓ | | ✓ | | ✓ | | |
| | CLMBR [21] | | | ✓ | | | ✓ | |
| | MedTPP [3] | ✓ | | ✓ | | ✓ | | |
| | MedGPT [7] | ✓ | | ✓ | | ✓ | | |
| Contrastive | OCP [1] | NA | NA | NA | NA | ✓ | | |
| | Hi-BEHRT [10] | NA | NA | NA | NA | | | |
| Masking | Med-BERT [16] | NA | | ✓ | | | ✓ | |
| | BEHRT [11] | NA | | ✓ | | | ✓ | |
| | Graph-Transformer [14] | NA | ✓ | ✓ | | | ✓ | |
| | EHR-PT [12] | NA | ✓ | ✓ | | | ✓ | |
| | RAPT [18] | NA | ✓ | ✓ | | | ✓ | |
| | CEHR-BERT [13] | NA | | ✓ | | | ✓ | |
| | GRACE [17] | NA | ✓ | ✓ | | | ✓ | |
| | CMS LDS BERT [8] | NA | | ✓ | | | ✓ | |
| | MTL GPR [20] | NA | | ✓ | | | ✓ | |
| | T-BEHRT [15] | NA | | ✓ | | | ✓ | |
| Causal | **ESGPT (Ours)** | ✓ | ✓ | ✓ | ✓ | ✓ | ✓ | ✓ |

Table 5: A summary of existing foundation models over EHR data, broken down by what modalities of data they model and how they report evaluation results.

# 6 Full Architecture Details

These are documented fully at `https://eventstreamml.readthedocs.io/en/dev/usage.html#pre-built-models` as well.

## 6.1 Shared Components

Both our model varieties build upon the GPT-Neo pre-trained transformer code base[3] for core layer and module code. We augment these layers with custom input embedding layers, temporal position embeddings, and output layers that emit generative distributions spanning classification, regression, and time-to-event covariates. In the sub-sections below, we will describe each of these unique properties, and any other areas that deviate from a standard transformer, for both architectures.

## 6.2 Custom Input Embedding Layer

To efficiently embed ESGPT pytorch batch objects for generative modeling, we use an input embedding layer that produces temporal position embeddings and data embeddings, then combines them for input event/dependency graph element embeddings. Each of these capabilities are documented below.

**Data Embedding** The data embedding layer[4] uses a set of embedding bag operations to embed categorical codes and/or continuous values from an ESGPT `PytorchBatch` object. There are multiple modes it can be used, but the recommended usage (controlled by hyperparameters) builds on the embedding strategy presented in [4] in which categorical codes (e.g., laboratory tests) are embedded via one embedding layer, then through a second embedding layer those codes that have associated values are embedded again and scaled by their measured values (which are normalized to have zero mean and unit variance). Both results are then summed to form the final data embedding layer for each measurement, and all measurements within an event or dependency graph node are then summed together to produce the overall event or node data embedding.

**Temporal Embedding** To emulate position embeddings from NLP foundation models, we support both fixed frequency sinusoidal absolute temporal embeddings[5] or, building on the work of [23], learnable frequency sinusoidal absolute temporal embeddings.[6] The embeddings both apply sinusoidal functions of varying frequencies to the number of minutes since the start of the patient's sequence of each event to form a final embedding of the time-point of the event. Time embeddings are then either added to the whole event embedding (for conditionally independent models) or to each dependency graph element embedding (for nested attention models).

## 6.3 Core Transformer Blocks

### 6.3.1 Conditionally Independent Point-process Transformer

The conditionally independent point process transformer is identical to a standard GPT Neo-X transformer. Measurements are fully aggregated together within an event to form event embeddings, which are then processed via an autoregressive transformer in the normal way.

---

[3] `https://raw.githubusercontent.com/huggingface/transformers/e3cc4487fe66e03ec85970ea2db8e5fb34c455f4/src/transformers/models/gpt_neo/modeling_gpt_neo.py`

[4] Documented in `https://eventstreamml.readthedocs.io/en/dev/usage.html#data-embedding-dataembeddinglayer` and `https://eventstreamml.readthedocs.io/en/dev/api/EventStream.data.data_embedding_layer.html`

[5] `https://eventstreamml.readthedocs.io/en/dev/api/EventStream.transformer.transformer.html#EventStream.transformer.transformer.TemporalPositionEncoding`

[6] `https://eventstreamml.readthedocs.io/en/dev/api/EventStream.transformer.transformer.html#EventStream.transformer.transformer.LearnableFrequencySinusoidalTemporalPositionEncoding`

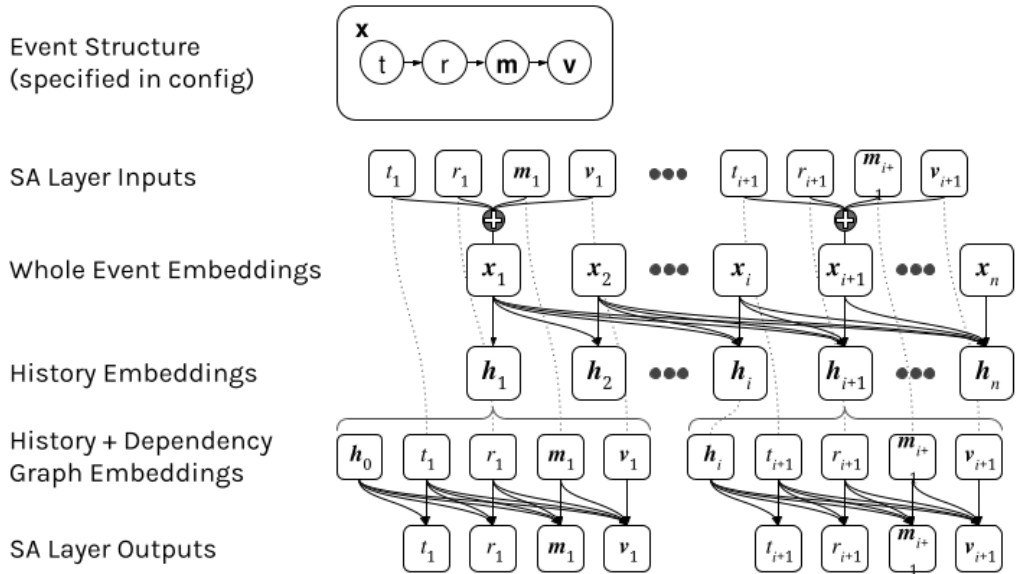

Figure 1: The nested attention layer, in visual form.

### 6.3.2 Nested Attention Transformer

The Nested attention transformer uses a two-stage attention mechanism to process data.[7] In this model, rather than whole event embeddings being produced by the input layer, instead, events are split into nodes within a pre-specified intra-event dependency graph, and whole node events are summarized from their component measurements. Then, within each layer, the model first produces a preliminary whole event embedding by pooling over the entire dependency graph, then it performs a historical self attention stack, then it uses the output of that stack as a new, "history" node at the beginning of the specified dependency graph, offset by one event so that each dependency graph node for a given event now has a node summarizing all events prior to that event. Then, the second stage of attention is a self attention layer over this dependency graph, summarizing all prior dependency graph elements along the graph and all historical events. The output of this second stage is the output of the "attention" component of each layer of the Nested Attention transformer. In this way, the model iteratively processes whole events and dependency graphs, ensuring appropriate causal flow to reflect intra-event dependencies. See Figure 1 for a visual representation.

### 6.4 Output layers

Naturally, ESGPT models also differ from canonical NLP LMs in their output layers. Whereas NLP GPTs use categorical prediction heads across the input vocabulary, ESGPT models must support three separate layer types. These include time-to-event prediction heads, for which we currently support an exponential distribution head and a mixture-of-lognormals distribution heads, building upon the work in [19], categorical prediction heads, using multinomial output distributions, and continuous value prediction heads, where the model predicts a mean and variance output which is then used to craft a re-parametrizable distribution output for probabilistic prediction. Using these output distributions (which for nested attention models are applied at a per-dependency graph node level, for predicting only the properties present in the subsequent dependency graph node) are used during generative pre-training by providing likelihood estimates of the true observed data, with which the model can then be trained by minimizing negative log likelihood, much as language models are trained in NLP.

---

[7]https://eventstreamml.readthedocs.io/en/dev/usage.html#nested-attention-point-process-transformer

## Supplementary References