# OpenReview forum: "Event Stream GPT: A Data Pre-processing and Modeling Library for Generative, Pre-trained Transformers over Continuous-time Sequences of Complex Events"
_NeurIPS.cc/2023/Track/Datasets_and_Benchmarks — NeurIPS 2023 Datasets and Benchmarks Poster_

### Official Review · Reviewer_17F7 · 2023-06-24
**Event Stream GPT: A Data Pre-processing and Modeling Library for Generative, Pre-trained Transformers over Continuous-time Sequences of Complex Events**

**Rating:** 7
**Confidence:** 3

**Strengths:**

Versatility Extension: The paper addresses the potential of GPT models beyond NLP by enabling their application to continuous-time sequences of complex events, specifically medical record datasets. This extension expands the utility and versatility of GPTs.

Streamlined Process: ESGPT provides a comprehensive library that streamlines the end-to-end process of building GPTs for event sequences, offering simplified dataset construction, a compatible modeling API, and standardized evaluation procedures. This makes it easier for researchers and practitioners to utilize GPTs in these domains.

Open-Source Contribution: The paper introduces ESGPT as an open-source library, allowing for widespread access and fostering collaboration among researchers and developers.

**Additional Feedback:**

Providing more details on the standardized evaluation processes and how they assess few and zero-shot performance of pre-trained models would enhance the paper's practical relevance.
Considering the open-source nature of ESGPT, it would be useful to discuss the availability of documentation, code examples, and community support for users to effectively utilize the library.

**Clarity:**

The paper is generally well-written and effectively presents the purpose, methodology, and contributions of ESGPT. The concepts are explained clearly, and the accompanying examples aid in understanding the proposed framework.

**Correctness:**

The claims made in the submission appear to be correct, given the information provided. However, without a thorough evaluation or benchmarking, it is difficult to verify the performance and effectiveness of ESGPT on some specific domains.

**Documentation:**

As the submission introduces a software utility, it would be beneficial to provide sufficient detail on data collection, organization, availability, maintenance, and ethical considerations. This information would ensure reproducibility and help users understand the dataset's characteristics and limitations.

**Ethics:**

None.

**Limitations:**

Yes

**Opportunities For Improvement:**

Narrow Application Scope: The proposed framework focuses specifically on continuous-time sequences of complex events, limiting its applicability to other types of data or domains. It would be beneficial to discuss potential extensions or adaptations of ESGPT to different contexts.

Evaluation and Benchmarking: While the paper mentions standardized evaluation processes, it does not provide a detailed analysis of the performance of ESGPT on specific tasks or benchmarks. Including such evaluations would strengthen the paper's claims.


**Relation To Prior Work:**

The paper adequately discusses how ESGPT differs from previous contributions, specifically in extending the application of GPTs to continuous-time event sequences. However, a more comprehensive discussion and comparison with related works in the field of event sequence modeling would provide further context and highlight the novelty of ESGPT.

**Summary And Contributions:**

The paper introduces Event Stream GPT (ESGPT), an open-source library designed to extend the applicability of generative pre-trained transformers (GPTs) beyond natural language processing (NLP) to continuous-time sequences of complex events, such as medical record datasets. ESGPT streamlines the process of building GPTs for event sequences by providing flexible dataset construction, a modeling API for GPTs with intra-event causal dependency structures, and standardized evaluation processes for fine-tuning tasks. The primary contribution of the paper is the introduction of ESGPT, which fills the gap in suitable tools for constructing and evaluating GPT models for continuous-time event sequences.

---

> ### Author Response · Authors · 2023-08-22
> **Review Response**
>
> Thank you for your insightful recommendations; we are very glad to see that you liked the paper! Below, we provide detailed responses to each of your concerns; if, after these updates, you don’t feel that your concerns have been sufficiently addressed or you have other feedback to offer, we would be very grateful for any further commentary or recommendations to best improve this work. Thank you again!
>
> ### Concern 1: Comment on Extensibility to Additional Modalities
>   > The proposed framework focuses specifically on continuous-time sequences of complex events, limiting its applicability to other types of data or domains. It would be beneficial to discuss potential extensions or adaptations of ESGPT to different contexts.
>
> This is a very good suggestion; we have added additional content in the future work (section 6) discussing the extensibility to additional modalities or contexts, particularly waveform or wearables data.
>
> ### Concern 2: Evaluation and Benchmarking
>   > Evaluation and Benchmarking: While the paper mentions standardized evaluation processes, it does not provide a detailed analysis of the performance of ESGPT on specific tasks or benchmarks. Including such evaluations would strengthen the paper's claims.
>
> This is also an excellent point; we have added additional content in section 3 and in the SI comparing ESGPT to the most comparable existing pipelines on measures of computational performance. As ESGPT’s primary value is as a data pre-processing/representation tool and as a set of modeling APIs, rather than in providing pre-trained models or concrete architectures directly, assessing the computational performance of the pipeline is the appropriate empirical comparison, rather than raw model performance. That said, we have also greatly expanded and promoted to the main body (in Section 5.4) content that demonstrates ESGPT’s usage over the MIMIC-IV dataset and the resulting performance therein. Producing a formal benchmark of the provided sample architectures within ESGPT on a set of benchmark tasks is also on our future roadmap, and is noted in Section 6.
>
> ### Concern 3: Related Works
>   > However, a more comprehensive discussion and comparison with related works in the field of event sequence modeling would provide further context and highlight the novelty of ESGPT.
>
> Thank you for your suggestion! We would welcome any suggestions of additional related literature to include among our comparisons; in the interim, we have added several additional references to the work focusing on alternate modalities and evaluation questions relating to ESGPT style models, in addition to our existing related literature on comparison pipelines, existing model architectures, and related paradigms such as temporal point processes (Section 6).
>
> ### Concern 4: Additional details on data collection, organization, availability, maintenance, and ethics
>   > As the submission introduces a software utility, it would be beneficial to provide sufficient detail on data collection, organization, availability, maintenance, and ethical considerations. This information would ensure reproducibility and help users understand the dataset's characteristics and limitations.
>
> This is a great suggestion; appropriate commentary on these points is always warranted. That said, we do want to clarify that ESGPT does not, in and of itself, release any dataset. Rather, it is a tool for usage of other datasets that users may have access to locally, to better enable reproducible research and effective use of that data. That being said, we do feel more commentary on these concerns is still appropriate, so we have added content to our main text highlighting possible ethical risks and limitations relating to dataset usage in Section 6. Please let us know if we have misunderstood your concern or these additions do not remedy it.
>
> ### Concern 5: Documentation and Evaluation Processes Details
>   > Providing more details on the standardized evaluation processes and how they assess few and zero-shot performance of pre-trained models would enhance the paper's practical relevance. Considering the open-source nature of ESGPT, it would be useful to discuss the availability of documentation, code examples, and community support for users to effectively utilize the library.
>
> Thank you for raising this point of confusion! We’ve added a pointer to our extensive documentation (https://eventstreamml.readthedocs.io/en/dev/overview.html), which also points to github issues as our support vehicle, in the Introduction. The paper has also been extended via Section 5.4 and the documentation extended here as well (https://eventstreamml.readthedocs.io/en/dev/usage.html#fine-tuning) to better highlight few and zero-shot evaluation results. Full examples of scripts that can be used to run such models can be found in the MIMIC-IV tutorial, here: https://eventstreamml.readthedocs.io/en/dev/MIMIC_IV_tutorial/index.html
> Please let us know if this area is still unclear.

---

### Official Review · Reviewer_NNj7 · 2023-07-18
**A sequential electronic healthcare records pre-processing framework augmented by large language models**

**Rating:** 4
**Confidence:** 5

**Strengths:**

- This paper establishes an advanced framework aimed at extracting Electronic Health Records (EHRs) to structure sequential event inputs, which can then be used to construct generative Language Models (LLMs).

- Comprehensive documentation of the software's APIs is provided in this paper.

**Additional Feedback:**

1. L1 "(GPTs, a.k.a. "Foundation Models")": it should be clarified that GPTs belong to the conception of Foundation Models but they are not identical.

2. Sec. 1: reading through the abstract and introduction section, it is still not quite clear which tasks this paper proposes to resolve by GPT models. It is recommended to put more descriptions of the tasks at the beginning of this paper. In Sec. 2, it is presented that GPTs are for modeling sequential medical event data. Nonetheless, it is still elusive about the conception of "modeling" and how we leverage this modeling capability for specific tasks, e.g., health outcome prediction, treatment recommendation, etc.

3. "Fig. 1: Figure 1: The general data model of the EFGPT data pipeline." in Appendix, the figure seems to be missing.

**Clarity:**

Sort of, though, there is still a space for further improvement of the presentation.

**Correctness:**

As far as I am concerned, there is no experimental design and results presented in the paper.

**Documentation:**

This paper offers rather detailed documentation with around 200 pages in the appendix.

**Ethics:**

It is suggested that the authors highlight the ethical concerns when using pre-trained GPT models, e.g., ChatGPT, on medical datasets, because it may raise privacy concerns.

**Limitations:**

Please refer to the section "*Opportunities For Improvement".

**Opportunities For Improvement:**

- The complexity of the paper, coupled with a somewhat misleading title, may cause some confusion for readers. The title suggests that the main objective of the proposed framework is to utilize or train GPT models for medical predictive tasks. It's not until Section 3 that a clearer understanding of the authors' work and specific tasks is obtained.

- The paper does not provide enough quantitative results for a substantial comparison with preceding methods. Without this, it is difficult for readers to grasp the range of machine learning tasks the proposed framework is capable of solving, as well as the methodology for achieving these tasks.

**Relation To Prior Work:**

A clear discussion of the difference from prior arts is offered.

**Summary And Contributions:**

This paper introduces a framework for pre-processing, training, and predicting based on sequential electronic healthcare records, specifically designed for large language models:

- It outlines the process of constructing input datasets using configuration files.

- It utilizes the HuggingFace Transformers API to either access pre-trained models or locally train proprietary models.

- It includes an evaluation pipeline for assessing the performance of trained models or examining the zero-shot capabilities of these models.

---

> ### Author Response · Authors · 2023-08-22
> **Review Response**
>
> Thank you for your clear, constructive, and excellent feedback! We greatly appreciate your time, and have worked to address all of your concerns and better demonstrate the value of our approach. We detail the concerns you raise and our responses below; if, after these updates, you still feel that our work is not ready for acceptance, we would love to hear additional feedback from you on how we can make the work better, either for further revisions here or future submissions. Thank you again!
>
> ### Concern 1: Unclear Problem Framing
>   > The complexity of the paper, coupled with a somewhat misleading title, may cause some confusion for readers. The title suggests that the main objective of the proposed framework is to utilize or train GPT models for medical predictive tasks. It's not until Section 3 that a clearer understanding of the authors' work and specific tasks is obtained.
>   > ...
>   > L1 "(GPTs, a.k.a. "Foundation Models")": it should be clarified that GPTs belong to the conception of Foundation Models but they are not identical.
>   > Sec. 1: reading through the abstract and introduction section, it is still not quite clear which tasks this paper proposes to resolve by GPT models. It is recommended to put more descriptions of the tasks at the beginning of this paper. In Sec. 2, it is presented that GPTs are for modeling sequential medical event data. Nonetheless, it is still elusive about the conception of "modeling" and how we leverage this modeling capability for specific tasks, e.g., health outcome prediction, treatment recommendation, etc.
>
> This is a great point of feedback for us, and we’ve worked hard to make the problem framing here clearer. In particular, we have added the below sentence to the Introduction of the paper to better clarify our problem set-up early on, and have also added examples of the ESGPT library in use over fine-tuning tasks on our MIMIC-IV example in Section 5.4 of the paper.
>
> New introduction sentence: “In particular, this paper presents a data pre-processing and modeling library that makes it significantly easier to build generative, auto-regressive transformers over event stream data, then to fine-tune or use those models for zero-shot prediction over diverse fine-tuning tasks on these data, such as in predicting the risk of a patient of dying in the hospital or of needing an early readmission to the hospital from structured EHR data.”
>
> We’ve also clarified the statement about GPTs vs. Foundation Models that you rightly flagged as being unclear, which now clarifies that GPTs are a type of Foundation Model.
>
> ### Concern 2: Missing Figure
>   > "Fig. 1: Figure 1: The general data model of the EFGPT data pipeline." in Appendix, the figure seems to be missing.
>
> We apologize; the supplement must have dropped the figure when we printed out the documentation from readthedocs (available here: https://eventstreamml.readthedocs.io/en/dev/) to pdf form. That figure should have been identical to Figure 1 in the main body, in any case, so it was not new content. Regardless, we should have clarified what the source for this figure was. We have also updated the supplementary material to point directly to the online documentation rather than containing a static PDF copy of the same, to solve this issue.
>
> ### Concern 3: Insufficient Model Performance
>   > The paper does not provide enough quantitative results for a substantial comparison with preceding methods. Without this, it is difficult for readers to grasp the range of machine learning tasks the proposed framework is capable of solving, as well as the methodology for achieving these tasks.
>
> Thank you for raising this issue. We would like to emphasize that this framework is not primarily about providing new models which achieve SOTA performance; rather, it is about enabling new modeling at scale on this modality by providing a standardized data extraction and pre-processing pipeline and a modeling API to leverage said data. Thus, our emphasis in the work is not around the performance of our proof-of-concept models, but rather around the viability of the framework on the whole to solve those core issues.
>
> That being said, we do feel that, as you and others have suggested, model performance should be better demonstrated here, and to that end we have added additional content clarifying the performance of our proof of concept models in section 5.4. We do not compare to preceding methods because (and this is part of the motivation for this framework) without a canonical data format, pre-processing system, and modeling API, such comparisons are very challenging and reproducibility is very limited. Additionally, there are no previously published examples (to the best of our knowledge) of models building fully generative temporal point processes on modalities of this complexity, so there are no direct comparisons we can make here, as noted in Section 5.2.

---

> > ### Comment · Reviewer_NNj7 · 2023-08-29
> >
> > I appreciate the authors' response and the edits made. However, I still believe it is important for benchmark paper to perform a comprehensive evaluation of diverse algorithms, or at least different GPTs in this case.

---

### Official Review · Reviewer_AZqA · 2023-07-19
**Review of Even Stream GPT**

**Rating:** 8
**Confidence:** 3
**Clarity:** The paper well-written and clearly st…

**Strengths:**

- The framework supports the expansion of GPTs to non-text applications, providing a principled pipeline for pre-processing and generation of arbitrary event streams.
- The software follows strong coding practices and is extensively documented.

**Additional Feedback:**

- The claim that 300,000 input patients from MIMIC-IV take up only 1.2 GB on disk seems a little exaggerated given that a) only 50,000 patients in MIMIC-IV have an ICU stay to begin with and b) only ~10,000/50,000 patients are actually included in the final extract due to the choice of inclusion/exclusion.
- The elements of PytorchBatch could be named clearer. From inspection of the instantiated object, I can deduce that “measurement_indices” correspond to the type of measurement (gender, lab_itemid, …) whereas the “indices” correspond to the categorical value of the measurement (”F”, “51214 (mg/dL)”, …). I would have not able to deduce this from the paper or documentation alone.
- The package currently relies on a very long list of dependencies, including some that are deprecated (e.g., both pytorch_lightning and lightning are listed), which may be reduced. Furthermore, while a full list of all dependencies including builds is good, it makes it hard to install on other platforms (e.g., on Mac for quick exploration). An environment description `--from-history` could help here.
- Minor typos in lines 99 and 101.

**Correctness:**

The pre-processing appears (overly) stringent. In the MIMIC IV example, 1,529,619 / 1,530,351 (99.9%) of Ketone measurements (itemid = 50947) are thrown out because they have no recorded measurement unit. The remaining 0.01% all have the same unit. Excluding samples with missing units may be the correct action in some cases but here it clearly leads to a tacit deletion of almost all data. Ketone is just an example and this likely happened for many other MIMIC data items due to the same or similar reasons. At a minimum, the data pipeline therefore needs to become fully transparent on when and why it throws away data.

**Documentation:**

There is a good use of docstrings and an extensive manual, including a working example on the openly available MIMIC IV database.

**Ethics:**

No ethical concerns.

**Limitations:**

The authors have properly addressed the technical limitations and possibilities for future work. The authors also mention the need for future evaluation metrics in the areas of fairness and privacy. Another potential (unmentioned) risk could be the increasingly opaqueness of processing MIMIC or similar EHRs as a whole without manual validation of data items and the consequent risk of introducing or propagating data bias etc. (see section on Correctness).

**Opportunities For Improvement:**

- MIMIC IV is used as an example in the paper and details/results are mentioned for pre-processing and modelling of intra-event dependencies. However, the evaluation results (which show negligible performance) are only available in the supplement. While I agree that the main focus of the paper should be the framework structure, I believe the model performance on MIMIC as well as a breakdown of which patients were left in the cohort should be included in the main body to clearly show its primary use as dummy data rather than a benchmark for future modelling.
- It is currently unclear if the NestedAttention model is a novel architecture proposed by the authors or if it is based on pre-existing work. The supplement is also relatively vague on the details of the model.

**Relation To Prior Work:**

The paper carefully considers the capabilities and shortcomings of existing preprocessing facilities and GPTs for electronic health records.

**Summary And Contributions:**

The paper introduces a framework for the training of Generative Pre-trained Transformers over the domain of continuous-time event streams. The framework allows for the efficient pre-processing of complex real-world datasets (as showcased on MIMIC IV), transforming the data into a format suitable for model training in Pytorch. The framework comes with two GPT architectures adapted for event streams and further provides facilities for the evaluation of few- and zero-shot prediction.

---

> ### Author Response · Authors · 2023-08-22
> **Review Response**
>
> Thank you for your thorough and constructive review, and we are glad to hear you see the value in this work! We further greatly appreciate your feedback and have worked to address them in our revision, as detailed below.  If any of your concerns remain unaddressed and you feel you still cannot recommend this paper for acceptance, please let us know and we will both work to correct any remaining issues and greatly appreciate any additional feedback.
>
> ### Concern 1: Clearer Inclusion on Modeling Performance
>   > MIMIC IV is used as an example in the paper and details/results are mentioned for pre-processing and modelling of intra-event dependencies. However, the evaluation results (which show negligible performance) are only available in the supplement. While I agree that the main focus of the paper should be the framework structure, I believe the model performance on MIMIC as well as a breakdown of which patients were left in the cohort should be included in the main body to clearly show its primary use as dummy data rather than a benchmark for future modelling.
>
> You’re right; we should promote the dummy results over MIMIC-IV to the main body. To that end, we have significantly revised that content and included greater details on experiments one can run and the results we obtain on the MIMIC-IV example in the main body, in Section 5.4
>
> ### Concern 2: Unclear Novelty of the Nested Attention Model
>   > It is currently unclear if the NestedAttention model is a novel architecture proposed by the authors or if it is based on pre-existing work. The supplement is also relatively vague on the details of the model.
>
> While the Nested Attention model builds on many pre-existing works, it is a novel architecture in that we don’t know of any other papers that use this precise architecture. As you note previously, the focus on this submission is on the ESGPT pipeline, not the NestedAttention architecture, which is why we didn’t emphasize its novelty more in the paper. It is just that in order to perform this kind of modeling at full, generative scale, a novel architecture was needed, and this was the simplest we could think of that captured intra-event dependencies.
> Regardless, we have made this clearer in the text, stating in Section 4.3 “While the `NestedAttention` model builds on pre-existing works, to the best of our knowledge this precise architecture is novel.”
> We have also additionally added _significant_ additional modeling details in the SI in section “Full Architecture Details”.

---

> > ### Author Response · Authors · 2023-08-22
> > **Continued Review Response**
> >
> > ### Concern 3: Dropped Data and Overstated Data Compression for MIMIC-IV
> >   > The pre-processing appears (overly) stringent. In the MIMIC IV example, 1,529,619 / 1,530,351 (99.9%) of Ketone measurements (itemid = 50947) are thrown out because they have no recorded measurement unit. The remaining 0.01% all have the same unit. Excluding samples with missing units may be the correct action in some cases but here it clearly leads to a tacit deletion of almost all data. Ketone is just an example and this likely happened for many other MIMIC data items due to the same or similar reasons. At a minimum, the data pipeline therefore needs to become fully transparent on when and why it throws away data.
> >   > ...
> >   > The claim that 300,000 input patients from MIMIC-IV take up only 1.2 GB on disk seems a little exaggerated given that a) only 50,000 patients in MIMIC-IV have an ICU stay to begin with and b) only ~10,000/50,000 patients are actually included in the final extract due to the choice of inclusion/exclusion.
> >
> > You’re right that there was a lack of transparency here, and that we unintentionally overstated the magnitude of the storage savings. We’ve taken steps to address both of those issues, and would greatly welcome further feedback. In particular,
> >   1. We’ve extended the MIMIC-IV tutorial in our documentation to note the impact of various pre-processing steps and data configuration files to note why and what data is discarded, here: https://eventstreamml.readthedocs.io/en/dev/MIMIC_IV_tutorial/index.html#dropped-data.
> >   2. We’ve revised the computational performance comparisons section to better reflect the impact various stages of pre-processing and the pipeline have on the final numbers, in Section 3.
> >
> > In particular, in section 3, we now state:
> >   > For our working example, our entire extraction and pre-procesisng pipeline over the MIMIC-IV dataset (yielding a cohort of approximately 12 thousand patients) takes only approximately 30 minutes. We further assess the performance of this pipeline on two other datasets on different compute environments, with cohort sizes of 116 thousand and 145 thousand patients, and find runtimes of 25 minutes and 62 minutes, respectively, further demonstrating the strong performance of this pipeline. Output dataset sizes are also universally small, with no dataset exceeding 4GB of final space on disk, with only standard compression.
> >
> > We further improve the utility of these comparisons by adding a proxy comparison to the omop-learn pipeline, stating the following (also in Section 3):
> >   > Unfortunately, there are no direct competitors that replicate all the steps of ESGPT in a consistent manner against which we can compare these results. However, to contextualize them, in order to run omop-learn on MIMIC-IV, one would first need to convert MIMIC-IV to the OMOP format, a process that is supported by existing ETLs. Running only these existing ETLs requires a runtime of over 37 minutes (https://github.com/OHDSI/MIMIC/blob/df97a75cd974c491e595c8b007a79f7326066cb1/z_more/run_times.txt#L4), meaning that _the entire ESGPT data pre-processing pipeline, which runs from scratch on MIMIC-IV and includes extraction, pre-processing, and final representation, all happens faster than just the initial extraction and data conversion stage required for omop-learn_ (though, naturally, these comparisons were run on different systems, so this is not a direct comparison).
> >
> > ### Concern 4: `PytorchBatch` Documentation:
> >   > The elements of PytorchBatch could be named clearer. From inspection of the instantiated object, I can deduce that “measurement_indices” correspond to the type of measurement (gender, lab_itemid, …) whereas the “indices” correspond to the categorical value of the measurement (”F”, “51214 (mg/dL)”, …). I would have not able to deduce this from the paper or documentation alone.
> >
> > Thank you for pointing this out! We’ve extended the documentation in the source code (https://eventstreamml.readthedocs.io/en/dev/api/EventStream.data.types.html#EventStream.data.types.PytorchBatch) to make these facets clearer. We’ve also added more details that are not dependent on MIMIC access to the new, synthetic data tutorial, here (https://eventstreamml.readthedocs.io/en/dev/_collections/local_tutorial_notebook.html) to cover this point.

---

> > > ### Author Response · Authors · 2023-08-22
> > > **Continued Review Response**
> > >
> > > ### Concern 5: Dependency Management
> > >   > The package currently relies on a very long list of dependencies, including some that are deprecated (e.g., both pytorch_lightning and lightning are listed), which may be reduced. Furthermore, while a full list of all dependencies including builds is good, it makes it hard to install on other platforms (e.g., on Mac for quick exploration). An environment description --from-history could help here.
> > >
> > > Yes, you’re right, the dependency management in our repository was rather sub-optimal, so thank you for catching this. We’ve taken steps to simplify this and to make it much easier to install the package in general. The updated instructions to do so can be found in the `README` of the repository (currently on the dev branch) and in the associated usage guide on the online documentation, and only requires the user to run `pip install -e .` in the root directory of the repository. The conda yml files are included for posterity, but a minimum set of required dependencies is listed via the poetry tool in the `pyproject.toml` file.
> > >
> > > ### Typos
> > > Thank you for catching the typos in line 99 and 101! They have been corrected.

---

> > > > ### Comment · Reviewer_AZqA · 2023-08-29
> > > >
> > > > I have reviewed the revised version of your manuscript and would like to express my appreciation for the efforts you have made in addressing the comments and suggestions provided. From my point of view, the contributions of EventStream GPT have been further clarified and I am modifying my score accordingly.

---

### Official Review · Reviewer_VUXc · 2023-07-20
**This paper succinctly and clearly describes Event Stream GPT, a framework for building generative models over event stream EHR data in MIMIC IV - a timely and important contribution.**

**Rating:** 9
**Confidence:** 4
**Correctness:** The claims made in the submission are…

**Strengths:**

The authors describe three categories of contributions, describing the problem, current state of the art, and contributions of ESGPT for each:

1. Data Extraction and Pre-processing - ESGPT is designed for extraction and pre-processing of MIMIC-IV, is optimized for deep learning, and is designed to build a PyTorch dataset, data loader and embedding layer; configuration is specified via a YAML file, and an example is provided in the appendix
2. Building GPTs - ESGPT is designed to produce a generative model that supports modeling of historical dependencies, process of event sequences taking timestamps into account, and to construct time-dependent features (such as age) during generation
3. Evaluating Foundation Models - ESGPT supports unsupervised zero-shot evaluations, and has a PyTorch Lightning module for evaluation (including support for evaluation during hyperparameter tuning)


**Additional Feedback:**

There is a typo on page 3: "structued dataset" should read "structured dataset"


**Clarity:**

Yes, the paper is very well written - concise and nicely structured, breaking the contributions out into three areas and presenting the problem, current state of the art, and ways the ESGPT improves upon it for each.

**Documentation:**

Yes, the authors provide sufficient detail on the dataset used, extensive documentation in the 200+ page supplementary information, and code to try out ESGPT to demonstrate the contributions summarized above.

**Ethics:**

No, I have no ethical concerns with the paper.


**Limitations:**

The work presented is very wide reaching in its contributions and potential applications; while the authors acknowledge the limitations above, they do not propose mitigation strategies or future work to address them - such an addition would strengthen an already strong manuscript.

**Opportunities For Improvement:**

In the category of Evaluating Foundation Models, the authors list three areas that are "critical points to assess in early stage foundation models" but which are not explored in this manuscript:

1. Practical utility vs. current ML systems
2. Performance disparities across subject sub-populations
2. Identifiability of subject-level data from pre-trained model parameters

These are certainly important, and feasible (though potentially challenging) to assess in future work. Other limitations acknowledged by the authors are the lack of support for unit conversion, ontological aggregation and other application specific pre-processing.

**Relation To Prior Work:**

Yes, for each contribution the paper lays out the current state of the art, and how ESGPT differs. The authors also summarize related work in Tables 2 and 3, in an in-depth summary in the Supplementary Information, and in a more general dedicated section at the end of the manuscript.

**Summary And Contributions:**

In this paper, the authors present Event Stream GPT (ESGPT) - "an open source software package, API, and evaluation utility for foundation models over event stream data. ESGPT can represent diverse datasets across various sources in a unified manner, pre-process very large datasets extremely quickly through its use of the Polars library [26], and can Hyperparameter tune, pre-train, fine-tune, and perform zero-shot evaluation for foundation models through a Hugging Face compatible API." The paper describes the use of ESGPT for processing and building a generative model over MIMIC-IV data, which consists primarily of EHR data for ICU patients.

---

> ### Author Response · Authors · 2023-08-22
> **Review Response**
>
> Thank you so much for your insightful and clear feedback! We are very glad you see the value in this work. We have addressed your comments in our revision, as detailed below.
>
> ### Concern 1: Limitations and Mitigations Section
>   > The work presented is very wide reaching in its contributions and potential applications; while the authors acknowledge the limitations above, they do not propose mitigation strategies or future work to address them - such an addition would strengthen an already strong manuscript.
>
> Your suggestion here is excellent; to accommodate this, we have significantly expanded our commentary on the limitations of this pipeline, and on steps to take to address these limitations. This is reflected in the new manuscript, in Section 6, where we add some brief comments highlighting the importance of these issues in future work.
>
> ### Typo:
> Thank you for catching the typo on page 3! That has been corrected.

---

### Official Review · Reviewer_1o4p · 2023-07-22
**Promising approach that lacks a clear evaluation framework**

**Rating:** 4
**Confidence:** 3
**Clarity:** The paper is well-written and easy to…

**Strengths:**

Time series data (like EHR, Smartwatches) come from multiple modalities and having a unified approach to model such datasets can help in downstream tasks. Despite a lot of advances in vision and language foundation models, time series foundation models are required because most mobile sensing studies tend to have a smaller set of participants and downstream evaluation from large foundation models through few shot and zero shot learning will be very impactful for the community. However, this paper only provides an approach and lacks extensive comparison across multiple datasets or models. It also does not contribute any new dataset.

**Additional Feedback:**

In order to highlight multiple modalities it would be great to have support for datasets of other kinds like Empatica (Similar to the flirt library) or a broader set of algorithmic comparisons like Neurokit2.

1. Föll, Simon, et al. "FLIRT: A feature generation toolkit for wearable data." Computer Methods and Programs in Biomedicine 212 (2021): 106461.
2. Makowski, Dominique, et al. "NeuroKit2: A Python toolbox for neurophysiological signal processing." Behavior research methods (2021): 1-8.

**Correctness:**

The paper shows only MIMIC-IV but lacks a systematic comparison across different benchmarks.

**Documentation:**

The code is well-documented provides a detailed tutorial for MIMIC-IV.

**Ethics:**

There are no major ethical concerns with the submission that warrant further discussion or review.

**Limitations:**

There are no potential negative societal impact of this work.

**Opportunities For Improvement:**

The paper does not provide any empirical comparison of EventStreamGPT over multiple datasets or algorithms. For example how do the two approaches Conditionally Independent Model and Nested Attention Model perform on a bunch of datasets? The code documentation is sufficient but it does not address any of the criteria for this track. There is no empirical comparison between the algorithms. It also lacks any objective criteria that it presented in Table 2 against the alternative software stacks that exist.

**Relation To Prior Work:**

The paper highlights existing libraries relevant to MIMIC-IV. It also qualitatively compares other foundation models but lacks any empirical comparison of the approaches presented in this paper with other datasets.

**Summary And Contributions:**

The paper provides an open source library for event stream data like electronic medical records. EHRs with continuous time sequences of complex events tend to require modeling multiple temporal events across multiple modalities and internal dependencies. The library provides a end to end pipeline from (built on top of well established HuggingFace API), multiple model architectures Conditionally Independent Model and Nested Attention Model to support both existing approaches and newer modeling approaches. Finally it also provides support for seamless generation in a zero-shot setting.

---

> ### Author Response · Authors · 2023-08-22
> **Review Response**
>
> Thank you for the thoughtful and clear feedback! We have provided an updated version of the paper that we believe can help address your concerns, as detailed below. If any of these concerns remain unsatisfactory to you and prevent you from recommending this paper for acceptance, please let us know and we will both work to correct any remaining issues and greatly appreciate your feedback.
>
> ### Concern 1: Lack of empirical comparisons regarding neural network architectures:
>   > The paper does not provide any empirical comparison of EventStreamGPT over multiple datasets or algorithms. For example how do the two approaches Conditionally Independent Model and Nested Attention Model perform on a bunch of datasets?
>   > ...
>   > There is no empirical comparison between the algorithms.
>
> This is a great suggestion for future work, but ultimately the efficacy of these specific algorithms is not the focus of this work. Rather, this work is on the tool and interface to produce architectures like this at scale and on novel datasets. In part, this tool is important because it is difficult to provide these empirical comparisons; to do so today, you would need to write your own independent data extraction, pre-processing, and representation strategies to replicate previously published models in this space, which greatly hinders reproducibility. ESGPT provides a standard that all models in this space can follow, thereby greatly increasing reproducibility.
>
> We have added additional commentary to this point in the future works section. We have also (in line with this and other comments) added more details regarding the use of these proof-of-concept pipelines over MIMIC-IV in Section 5.4.
>
> ### Concern 2: Lack of empirical comparisons regarding data pipeline performance:
>   > It also lacks any objective criteria that it presented in Table 2 against the alternative software stacks that exist.
>
> This is a great point. We’ve added additional numbers quantifying some of these comparisons in the text, in Section 3.3, Table 3, as well as in the Supplementary Information to accommodate this result. While we cannot provide full quantitative comparisons across all metrics in Table 2 (as these pipelines do not all have identical capabilities), we provide the following comparisons in the added text:
>   1. First, we note that in comparison to the ~30 minute end-to-end runtime for ESGPT extracting and pre-processing MIMIC-IV, just the pre-pipeline extraction ETL required for omop-learn takes over 37 minutes, according to public pipelines. Therefore, our entire pipeline, including extraction, pre-processing, and final representation manipulation and storage, runs faster than just one part of the extraction process included in the omop-learn pipeline for the same data source.
>   2. Second, we’ve added Table 3, which compares the performance of deep-learning datasets produced through ESGPT against reproduced approximations of TemporAI and omop-learn, two of the most competitive existing pipelines from Table 2. We see that ESGPT matches or exceeds the performance of both of these pipelines on storage, load time, memory cost, batch iteration time, and batch memory cost, in many cases quite significantly. In particular, whereas TemporAI batches require approximately 875 MB per batch and 0.6 seconds of iteration time, ESGPT requires only 67 MB and 0.4 seconds for equivalent data. Note that additional details about these comparisons, including the assumptions made in each, the fact that omop-learn’s improvement over ESGPT in terms of batch memory cost is entirely explained by additional, optional metadata that ESGPT includes by default in each batch, and the rationale why omop-learn and TemporAI perform as they do can be found in the SI.
>
> ### Concern 3: Unsuited for Track
>   > The code documentation is sufficient but it does not address any of the criteria for this track.
>
> We feel that this submission is suitable for the track as it is a new tool to work with data more effectively in AI. This fits naturally under this track’s call for papers, which states “This track welcomes all work on data-centric machine learning research... This includes... Data-centric AI methods and tools, e.g. to measure and improve data quality or utility”.
>
> ### Concern 4: Additional data modalities or algorithms
>   > In order to highlight multiple modalities it would be great to have support for datasets of other kinds like Empatica (Similar to the flirt library) or a broader set of algorithmic comparisons like Neurokit2.
>
> This is a great suggestion for future work, and we have added additional commentary to that effect in Section 6 and added citations to both of the recommended resources. Thank you for these pointers!

---

### Author Response · Authors · 2023-08-22
**Overall Summary of Revisions**

Firstly, we would like to thank all reviewers for their thoughtful and insightful feedback. We are very pleased to see that reviewers, in general, see the significant value of our work, stating that ESGPT is “... a timely and important contribution”, that ESGPT “...supports the expansion of GPTs to non-text applications, providing a principled pipeline for pre-processing and generation of arbitrary event streams... [and] follows strong coding practices and is extensively documented”, and that ESGPT “...makes it easier for researchers and practitioners to utilize GPTs in these [event stream] domains”.

To address the excellent suggestions for further improvement, we have made a number of revisions to the manuscript posted with this update (modified text is colored red so you can easily see what changed). While we will post individual comments to each reviewer addressing all concerns raised explicitly, here we will also summarize the major changes made in this manuscript:

  1. We have added more explicit, quantitative comparisons of the computational performance of the data pre-processing and representation aspects of ESGPT against those of TemporAI and omop-learn, showing that ESGPT provides a faster pre-processing pipeline and a deep learning representation that is faster and more memory efficient to load and comparable to or faster to iterate through and use on the GPU than either other pipeline (in particular requiring only 8% of the GPU memory that TemporAI’s batches do).
  2. We have extended our section detailing fine-tuning usage and sample model performance and moved it to the main body, demonstrating how one can quantify both baseline and fine-tuning performance via simple scripts provided with ESGPT, and demonstrating what the proof-of-concept models yield on the MIMIC-IV dataset. We have also improved the results of these proof-of-concept systems by extending hyperparameter tuning capabilities to fine-tuning settings as well, demonstrating models where fine-tuning performance actually does yield a modest improvement over from-scratch training even in this small, proof-of-concept setting.
  3. We have made various clarifications to the text regarding problem framing, relation to prior work, and usage instructions.
  4. We have made numerous other small improvements to the library (to support its current, active use by the community), including:
Improvements to the installation process, which now supports a simple pip install -e . installation.
  5. We have added of a much more hands-on, detailed tutorial of the library over a synthetic dataset that is distributed with the source code and can be run even on a small, local machine, accessible here: https://eventstreamml.readthedocs.io/en/dev/_collections/local_tutorial_notebook.html
  6. We have added hyperparameter tuning support for baseline and fine-tuning models, including a pipeline for producing baseline (e.g., scikit-learn) performance on fine-tuning tasks that can search over historical aggregation window sizes alongside standard scikit-learn pipeline parameters.
  7. We have introduced a variety of other small improvements and bug-fixes.
  8. We have made small other changes to the manuscript text to ensure new content fits within the allowed ten pages.

We would like to note that the changes noted in this comment and in the individual reviewer comments are currently reflected in our development branch of ESGPT, so that further changes suggested in this period can be safely incorporated prior to a larger update to the main branch to avoid disruption to users. These changes can still be fully inspected on the github, and their documentation is also live on our staging documentation build, for your review.

Relatedly, we have also updated the tutorial MIMIC-IV repository that is referenced in our documentation to reflect these changes; that repository is also updated such that the development branch there matches the development branch of ESGPT, to avoid any issue with mismatched versions therein.

Finally, we also note that we have removed the static PDF dump of our online documentation that was formerly appended to our additional supplementary information in the PDF; this documentation still exists and is accessible to users at https://eventstreamml.readthedocs.io/en/dev/, we have just removed the PDF copy from the revision here so there is only a single, reliable source for these information.

---

> ### Author Response · Authors · 2023-08-25
> **Further Experimental Update**
>
> Thank you again to all reviewers; we have just posted another minor revision that further expands and clarifies the proof-of-concept modeling results over MIMIC-IV, demonstrating that using this our pipeline end to end, we obtain models where fine-tuning performance exceeds networks trained from scratch with a lift in AUROC of 0.02 on our sample readmission risk prediction task and a lift in AUROC of 0.04 on our sample in hospital mortality prediction task. These results are still merely proof of concept results, and the primary value of this work is in the flexibility of the data processing pipeline and the set of unified modeling APIs enabling easy pre-training, fine-tuning, and from scratch evaluations, but we wanted to further clarify that using this pipeline now, one can obtain proof-of-concept results demonstrating the possible benefits of building models of this kind, further motivating the utility of our pipeline.

---

### Decision · Program_Chairs · 2023-09-22

**Decision:**

Accept (Poster)

**Comment:**

This paper introduces Event Stream GPT (ESGPT), an open-source library designed to extend the applicability of Generative, pre-trained transformers (GPTs) to continuous-time sequences of complex events such as medical record datasets. It also describes the use of ESGPT for building GPT models over the MIMIC-IV electronic health record dataset.

The paper is well-written and presents the methodology and contributions of ESGPT.
The authors have made several changes and clarifications to address the reviewers’ questions and comments.

The paper would benefit from adding an empirical comparison of ESGPT over multiple datasets and algorithms.